# Disentangling the Drivers of the Sampling Bias of Freshwater Fish across Europe

**Marta Rodríguez-Rey [1,2,*]** and **Gaël Grenouillet [1,3]**

1   Laboratoire Evolution et Diversité Biologique, UMR5174, Université de Toulouse III Paul Sabatier, CNRS, IRD, 31062 Toulouse, France
2   University of Alcalá, Faculty of Sciences, Life Sciences Department, FORECO Lab, Crta. A-2 Km. 33.6, 28805, Alcalá de Henares, Madrid, Spain
3   Institut Universitaire de France, 75005 Paris, France
*   Correspondence: marta.rodriguezrg@uah.es

**Abstract:** The Wallacean shortfall refers to the knowledge gap in biodiversity distributions. There is still limited knowledge for freshwater fish species despite the importance of focusing conservation efforts towards this group due to their alarming extinction risk and the increasing human pressure on freshwater ecosystems. Here, we addressed the Wallacean shortfall for freshwater fish faunas across Europe by using the completeness indicator derived from species accumulation curves to quantify the fish sampling efforts. The multiple potential drivers of completeness that were previously related to the sampling efforts for other species (i.e., population density, nature reserves, or distance to cities) were tested using a $10 \times 10$ km$^2$ grid resolution, as well as environmental (e.g., climatic) factors. Our results suggested that although there was an overall spatial pattern at the European level, the completeness was highly country-dependent. Accessibility parameters explained the sampling efforts, as for other taxa. Likewise, climate factors were related to survey completeness, possibly pointing to the river conditions required for fish sampling. The survey effort map we provide can be used to optimize future sampling, aiming at filling the data gaps in undersampled regions like the eastern European countries, as well as to account for the current bias in any ecological modeling using such data, with important implications for conservation and management.

**Keywords:** survey effort; Wallacean shortfall; completeness; SDM; accessibility; stream fish; reliability

## 1. Introduction

Since the last decade, the boost of big data has promoted the compilation of the spatial records of species in repositories which are ready to be downloaded and used for research purposes [1], easing the development of ecological and evolutionary studies without the need to conduct fieldwork. The available data come from natural history collections, national and international environmental agencies, and research institutions [2] and even from the volunteers that participate in citizen science programs by collecting biodiversity observations [3]. Such geographical data have been used in diverse types of research, ranging from evolutionary studies [4], the assessment of extinction risks [5], the modeling of species' responses to environmental changes, and the forecasting of species' range contractions or shifts under climate changes [6,7].

However, the biodiversity data have some quality limitations [8]. This limitation, known as the Wallacean shortfall, refers to the lack of information about species' geographical occurrence in some regions [9,10]. Despite the high increase in the number of spatial records [11,12], the biases are pervasive in the species databases commonly used for Species Distribution Model (SDM) building [13]; most distributional databases still have a sampling bias [14], and some taxa are still undersampled [15].

This bias can affect the type and explanatory power of the environmental factors and especially the spatial predictions [16,17], highly affecting decision making and therefore

conservation and management. The availability of spatial records highly impacts the results of the SDMs, which are methodologies that mainly rely on the distribution of the species to infer the patterns of extinctions [18], the expansion of invasive species [19,20], or the predictions of global change impacts [21]. For this reason, data sampling was considered a priority for the research and development of SDMs [22] because the sampling bias in the distribution records can generate distorted model predictions, thereby calling into question their reliability to inform policy and management decisions [17,23–25].

Under the ongoing global changes, elucidating the human factors impacting biodiversity and environmental processes is a priority [26]. If species distributions are biased towards those places with higher human activity just because the sampling has been more intense in those areas, then the models applied to management and policy will be erroneous, and therefore, resources may be allocated to the wrong conservation measures. Areas with higher accessibility, which are closer to infrastructures or have some degree of protection or attractiveness, have been reported as the main drivers of sampling bias [27–30]. For this reason, to avoid the confounding effect of sampling bias, there is an increasing awareness in the scientific community of the importance of correcting or controlling for this bias and, consequently, numerous approaches [31,32] and software packages [33,34] have been developed for this purpose. Yet, improving the knowledge about the Wallacean shortfall is considered a challenge for many taxa [35], such as fish [36].

The survey coverage is poorer for aquatic taxa than for terrestrial taxa, and freshwater species, especially invertebrates, are scarce in international catalogues such as GBIF [37]. Nonetheless, freshwater ecosystems are some of the most threatened ecosystems on Earth [38]; also threatened are the freshwater species [39] directly competing with humans for resources [28]. Due to the increasing human population demanding multiple resources from freshwater ecosystems, ranging from direct products such as water or fish to recreational ecosystem services [40], there is an increasing need for effective conservation approaches to freshwater fish, and this requires reliable data [41]. Therefore, investigating and filling the knowledge gap regarding the aquatic biodiversity data should be a priority. Few studies have focused on the sampling bias related to freshwater species, except a few cases focusing on beetles [42] and amphibians [28]. Pelayo-Villamil et al. [43] evaluated the sampling completeness of freshwater fish species at a global scale, using countries as the spatial resolution, and found that countries with relatively accurate inventories had regions or provinces where the accuracy was low. Furthermore, the drivers of completeness have been evaluated for multiple taxa [24,44–46], but there is still limited information about the factors affecting the freshwater fish sampling efforts [47].

In this study, we aimed to assess the sampling efforts in the different European ecoregions [48] for freshwater fish species and to investigate the drivers of inventory completeness and how they differ from the sampling bias drivers in terrestrial ecosystems to improve our knowledge about the data limitations in freshwater ecosystems, and we propose solutions to improve the freshwater fish biodiversity assessments.

## 2. Methods

### 2.1. Study Area and Spatial Records

The study area comprised the area covered by Great Britain and the Isle of Man, Spain, Portugal, France, Italy, Belgium, the Netherlands, Denmark, Norway, Germany, Sweden, Poland, Slovenia, Romania, Bulgaria, and Greece. These countries had available data on fish distribution and represented the whole European area with samples covering all the ecoregions for the rivers and lakes defined in Annex XI of the Water Framework Directive [48], apart from the Tundra, Iceland, and Irish ecoregions.

The distribution records at the species level for all the countries were extracted from the GBIF [49], Table S1, and the available national databases, such as those of the United Kingdom [2], Sweden [50], France [51], Spain, and Portugal [52]. The national databases from countries with low or no records in the GBIF were requested by the authors when there was some information regarding the existence of a database, as was the case with

Slovenia [53]; the data were granted to develop this study. For all countries, each register included "species name", "latitude", and "longitude" in the EPSG:4326—WGS 84 coordinate systems and the "sampling date" (year) of the record. The year was the most reliable temporal resolution for most countries. Having a higher temporal resolution meant a more conservative approach to the completeness calculation. We curated the database by eliminating records without geographic coordinates and by correcting misspellings. We merged taxonomic synonyms based on FishBase [54] and removed duplicates by rounding the latitude and longitude in degrees to three decimal places to avoid inconsistency between databases [55]. Records were binned in a $10 \times 10$ km grid cells as this has been widely used in fish modeling [56].

### 2.2. Completeness Calculation

To evaluate the sampling bias in the European freshwater registry, we calculated the completeness as it measures how well the biodiversity inventories capture the full assemblage of species that are expected to occur at a given location, and we thereby determined the areas with deficient information [57]. We used the *KnowBPolygon* function in the KnowBR library in the R package [58] to assess the survey completeness across Europe at a $10 \times 10$ km$^2$ grid resolution. The KnowBR calculations are based on the slopes of the species accumulation curves accounting for the relationship between the number of species and the total number of records. The accumulation curve was estimated using the exact estimator and the formula defined in [59] and was adjusted to the rational functions; see the details in [58] for each spatial unit (i.e., grid). Then, the obtained extrapolated asymptotic value was used to calculate the completeness, which corresponds to the percentage of inventoried species over the total number of predicted species. This approach has been widely applied to calculate the spatial survey efforts for multiple taxa, such as butterflies, beetles [60], birds [35], and plants [61,62], as well as fish [43].

### 2.3. Predictors and Statistical Analyses

We investigated the potential factors explaining completeness by considering the anthropogenic predictors related to accessibility. We included road density, population density, proximity to main cities and airports, the human development index (HDI) [63], and the human footprint [64]. The proximity to the sampling sites, the attractiveness of the area, or the proximity to main cities with universities or research centres (Table 1) have previously been reported as being responsible for the sampling intensity for other species [36,44,61,65,66]. Protected areas were also included as the percentage of nature reserve areas in each grid cell (Table 1) according to their importance in previous studies [47]. Most anthropogenic predictors were obtained from RiverATLAS version 1.0, which is the HydroATLAS version at a stream level, with a river resolution of 500 m [67].

We also included environmental variables to account for the potential environmental sampling bias or climatic bias [37,65], referred to as the uneven representation of key environmental gradients by the occurrence records [37]. Although some taxa can be associated with humans for natural reasons (e.g., providing shelter or resources), this seemed irrelevant for fish species as more species are native and do not rely on human infrastructures for their survival. For environmental variables, we used the 19 bioclimatic variables available in the CHELSA V1.2 database [68].

For both the environmental and human predictors of completeness, we extracted the mean values for all the variables (Table 1) in each grid cell using the zonal statistic tool in QGIS version Las Palmas [69].

We considered all the 19 bioclimatic predictors because we did not anticipate beforehand the effect of any of them in the completeness distribution, and we applied a principal component analysis (PCA) to reduce the number of climate variables [70]. We performed a normalized PCA using the *prcomp* function from the package *stats* version 4.2.0 in R [71]. For the remaining predictors, we calculated the Pearson correlation to exclude those that were highly correlated (i.e., $r \geq 0.75$). We employed Generalized Linear Mixed models

(GLMM) to test the relationship between the completeness and the predictors, considering the country as a random factor. We checked the pseudo $R^2$ of the resulting model using library *MuMIn* in R [72] and evaluated the spatial autocorrelation of the model residuals through a variogram using the *gstat* package in R.

**Table 1.** Predictor variables used to investigate completeness distribution in Europe.

| Variable | Description | Source |
|---|---|---|
| Airport | Euclidean distance to the closest airport | http://worldmap.harvard.edu |
| Cities | Euclidean distance to the closest major city | https://hub.arcgis.com/maps/esri::world-cities-1/ |
| Road | Road density | https://www.hydrosheds.org/page/hydroatlas |
| Population | Population density in 2010 | https://www.hydrosheds.org/page/hydroatlas |
| HDI | Human Development Index in 2015 [65] | https://www.hydrosheds.org/page/hydroatlas |
| HFT | Human Footprint [66] | https://www.hydrosheds.org/page/hydroatlas |
| Bio 1 | Annual Mean Temperature | https://chelsa-climate.org/bioclim/ |
| Bio 2 | Mean Diurnal Range (mean of monthly temp (max temp–min temp)) | https://chelsa-climate.org/bioclim/ |
| Bio 3 | Isothermality (BIO2/BIO7) (* 100) | https://chelsa-climate.org/bioclim/ |
| Bio 4 | Temperature Seasonality (standard deviation *100) | https://chelsa-climate.org/bioclim/ |
| Bio 5 | Max Temperature of Warmest Month | https://chelsa-climate.org/bioclim/ |
| Bio 6 | Min Temperature of Coldest Month | https://chelsa-climate.org/bioclim/ |
| Bio 7 | Temperature Annual Range (BIO5-BIO6) | https://chelsa-climate.org/bioclim/ |
| Bio 8 | Mean Temperature of Wettest Quarter | https://chelsa-climate.org/bioclim/ |
| Bio 9 | Mean Temperature of Driest Quarter | https://chelsa-climate.org/bioclim/ |
| Bio 10 | Mean Temperature of Warmest Quarter | https://chelsa-climate.org/bioclim/ |
| Bio 11 | Mean Temperature of Coldest Quarter | https://chelsa-climate.org/bioclim/ |
| Bio 12 | Annual Precipitation | https://chelsa-climate.org/bioclim/ |
| Bio 13 | Precipitation of Wettest Month | https://chelsa-climate.org/bioclim/ |
| Bio 14 | Precipitation of Driest Month | https://chelsa-climate.org/bioclim/ |
| Bio 15 | Precipitation Seasonality (Coefficient of Variation) | https://chelsa-climate.org/bioclim/ |
| Bio 16 | Precipitation of Wettest Quarter | https://chelsa-climate.org/bioclim/ |
| Bio 17 | Precipitation of Driest Quarter | https://chelsa-climate.org/bioclim/ |
| Bio 18 | Precipitation of Warmest Quarter | https://chelsa-climate.org/bioclim/ |
| Bio 19 | Precipitation of Coldest Quarter | https://chelsa-climate.org/bioclim/ |

## 3. Results

The final databases contained 1,721,923 records of freshwater fish species, with an average of 77 different species per country. The countries with the highest number of records were France, followed by the United Kingdom and the Netherlands (Table S2). In contrast, Bulgaria, Romania, and Greece were the countries with the lowest number of records. The country with the highest number of species was Italy, followed by Greece and Spain; Norway, the United Kingdom, and Belgium had a smaller number of fish species (Table S2).

The first two axes of the PCA on the climatic variables, accounting for 38.1% and 30.8% of the total variability, respectively, were subsequently used (Table S3). The first principal component (PC1) was negatively associated with the temperature variables (Bio5, Bio10, and Bio1) and positively associated with the precipitation variables (Bio17, Bio18 and Bio14), thus reflecting a gradient from warm and dry climates (negative values) to cold and humid climates (positive values) (Table S4). The second component (PC2) mainly reflected a gradient from the climates characterized by low temperature seasonality (Bio4), high precipitation (Bio12, Bio13, and Bio16), and cool conditions during the dry and cold seasons (Bio9 and Bio6), respectively (i.e., stable and humid climates and negative values) to high temperature seasonality and dry conditions during the cold season (i.e., more variable climates over the year and positive values) (Table S4). Road density was highly correlated with population density and the human footprint, with a Pearson correlation (r) of 0.77 for both. The road density and human footprint were also correlated with the HDI (r = 0.76); so, we removed the human footprint and road density from further analysis.

From the 40,966 grid cells composing the study area, we found that for the whole of Europe only 18% of the grid cells were well surveyed, with completeness values over 80%. Only 49 cells in the study area were fully complete (i.e., completeness = 100). Another 18% of the cells had completeness values between 50 and 80, and 10% of the cells had values smaller than 50 but higher than 0. Fifty-four percent of the grid cells had 0 completeness, which differed highly among the countries (Figure 1). The countries with, on average, the lowest completeness were Romania, Bulgaria, Greece, Poland, and Italy. The country with the highest average completeness in the whole country area was the Netherlands, followed by Denmark, the United Kingdom, Slovenia, and France (Figure 1). According to the GLMM, all the predictors except population density significantly ($p < 0.05$) explained the completeness at the European scale. Both climate PCA axes had a negative effect on completeness (estimate = $-0.1553$, sd = 0.03, $p < 0.001$ and estimate = $-0.1599$, sd = 0.02, $p < 0.001$), as did distance to the airport (estimate = $-0.1782$, sd = 0.01, $p < 0.001$), distance to cities (estimate = $-0.1372$, sd = 0.01, $p < 0.001$), and percentage of nature reserves (estimate = $-0.0480$, sd = 0.04, $p = <0.001$), while the HDI had a positive effect (estimate = 0.1850, sd = 0.01, $p < 0.001$). The correlation of the data and the model residuals had low variation over distance (Figure S1). The conditional $R^2$, including the fixed effect and the random effect, was 0.64, whereas the marginal $R^2$ was 0.02, revealing that country, included as a random factor, explained most of the variance in completeness.

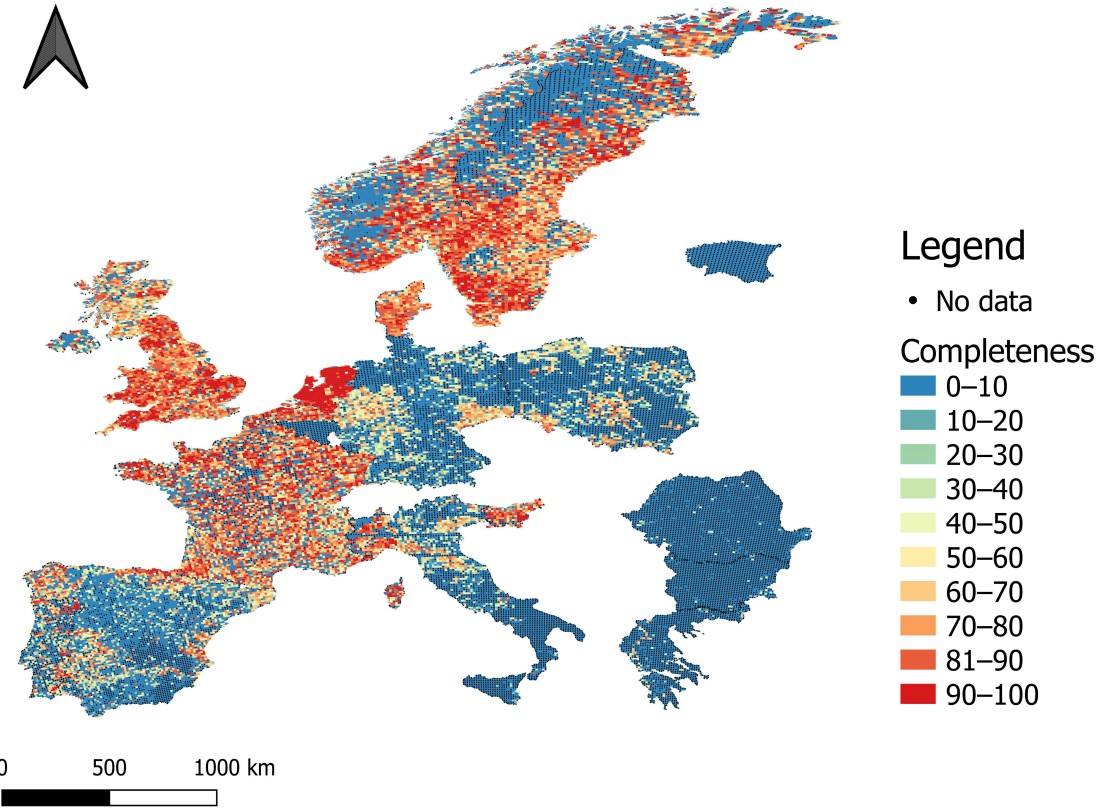

**Figure 1.** Completeness values for each $10 \times 10$ km$^2$ grid cell in Europe.

At a national scale, we found no significant predictors explaining the completeness of four countries (France, Bulgaria, Denmark, and Romania). From the remaining twelve countries, the distance to the cities and the percentage of nature reserves significantly influenced the completeness in eight countries (Table 2). For the distance to the cities, six countries had negative estimates, whereas two of them had positive estimates. For the nature reserves, four countries had positive estimates and four had negative ones. Both climate factors were significant for seven countries, with a positive effect, except for PC1 in Poland, which showed a negative effect.

**Table 2.** Estimates from the Generalized Linear Models (GLM) for the predictors significantly ($p < 0.05$) explaining completeness at the national level.

| | Belgium | Germany | Greece | Italy | Netherlands | Norway | Poland | Portugal | Slovenia | Spain | Sweden | United Kingdom |
|---|---|---|---|---|---|---|---|---|---|---|---|---|
| Climate PC1 | −2.7456 | −0.1909 | - | - | - | −1.1168 | 0.2427 | - | −1.4714 | - | −0.2945 | −0.4970 |
| Climate PC2 | - | −0.1945 | −0.4574 | - | - | −0.8693 | - | - | −0.8778 | −0.6246 | −0.3006 | −0.2778 |
| Nature Reserves | - | - | 0.2608 | −0.4622 | −0.2873 | −0.1568 | 0.1158 | - | - | - | −0.3434 | 0.1261 |
| Distance to airports | - | - | −0.3652 | −0.1318 | | | - | - | - | 0.1438 | −0.3290 | - |
| Distance to big cities | - | −0.1938 | - | −0.8213 | 0.4747 | −0.1593 | −0.3428 | 0.2814 | - | −0.2493 | - | −0.1686 |
| Population density | - | 0.2222 | - | −0.4364 | - | - | - | - | - | - | 0.0896 | - |
| HDI | 0.7212 | - | - | 0.6697 | - | - | −0.1088 | −0.2542 | 0.6577 | 0.1345 | −0.2426 | - |

## 4. Discussion

There are some countries in Europe, such as Bulgaria, Romania, Greece, and Estonia, with limited or inaccessible geographical information on freshwater fish. Eastern Europe was the region with a lower number of records as well as lower values of completeness with regard to terrestrial vertebrates [14]. Improving the data availability is especially important in the east of Europe considering that the best conserved freshwater areas are located in those countries [73]. A recent study [74], forecasted a strong decrease in species richness in eastern European countries, whereas an increase in western Europe is expected under different scenarios of climate change. Therefore, for an effective calculation of any extinction risk or niche shift experienced by freshwater species in those countries, there is a crucial need for additional registers through the promotion of data availability and data mobilization or the investment in a higher sampling effort in those regions with poor data. This goal can be achieved using different approaches, which are discussed further in the following paragraphs, to detect and document new species distribution with less effort and expense [75].

Fish species are more difficult to sample than terrestrial species, and the sampling requires expensive equipment and resources [76]. Citizen science has proven to be a highly useful tool for increasing the biodiversity databases and survey completeness [35,77]. In that context, anglers represent the group of people most likely to support this activity as freshwater fish are difficult for common citizens to spot [78]. Other recently developed techniques, such as eDNA, are showing a high ability for biodiversity assessments in rivers where there was previously no information [79,80]. Sometimes, the data that have been retrieved by researchers or environmental agencies for descriptive studies have never been published elsewhere [81]. Mobilizing these data from personal databases or research centres and museums might provide additional information [82]. Some data might still be undigitized [66]; so, funding for digitization might be quite useful but should be focused on the undersampled countries as it has been shown that the effect of digitization is less relevant in the well-sampled regions [61].

According to our study, completeness was country-dependent. The lack of information in some regions is relatively common and has been observed in other areas and at the country scale [81]. Our results highlighted the impact of political boundaries on the bias and the national or regional efforts in the biodiversity databases, which will affect the models and predictions applied to management and conservation. For instance, Titley et al. [83] suggested mapping transboundary range shifts globally to minimize biodiversity loss under global change. We agree that transboundary conservation efforts are key, but the differences in the completeness values and the bias in the data of the different countries might result in erroneous models. For that reason, modeling at the country level and then accounting for the results of the multiple countries might be a valid solution until a unified and unbiased database is available.

Globally, completeness was affected by all the considered factors except for population density, which only affected the survey quality in Germany, Sweden, and Italy. In the case of Italy, we found an unexpected pattern as the completeness was higher in the less populated areas, in contrast with other studies [44,84–86]. Our results had important

implications for research as those studies aiming to disentangle the anthropogenic factors driving the freshwater species' range expansion or contraction [87] or the fish biodiversity patterns [88] might conclude that the indicators of human impact are affecting freshwater fish, when the real meaning is that the areas close to cities or with a large human presence have a higher amount of data due to more intense sampling. However, as stated before, the global European pattern needs to be considered with caution because it is highly country-dependent. For instance, completeness increased in localities close to big cities, highlighting accessibility as a driver of bias in freshwater ecosystems or the presence of research centres in cities [46]. However, in Portugal, completeness was higher further from the cities and from areas with high human development, meaning that a more intense sampling was carried out in pristine and remote places. In addition, the gaps in fish inventories were located in areas with a low density of nature reserves [85], but the opposite was observed for those countries with high average inventory completeness, indicating that an overall substantial sampling effort in the whole country reduced the bias towards protected areas.

As in our study, the local conditions and the environment explained the completeness for different taxa of plants [89,90], amphibians, reptiles, birds [90], and different insects [46,91]. Climate influenced completeness in all the countries, with higher completeness in areas with high precipitation in the winter months and in areas where temperature seasonality and range were low. Moreover, completeness was higher in areas with high annual mean temperature and high temperature in the warmest quarter and low precipitation in the warmest and driest quarter/month. This may be a consequence of where aquatic habitats are most prevalent, as noted by Troia and McManamay [37], or might reflect the conditions of permanent rivers with continuous flow over the year, where the conditions are optimum for fish reproduction, or areas where the winter precipitation and the lack of extreme temperatures ensure a permanent stream flow. Sampling is also more frequent in the most common environments of a region [92]; so, the climates of those ecosystems and environments might be overrepresented. Another reason for the importance of the climate in fish completeness might be the environment or the ecosystems associated with the climate conditions.

To our knowledge, this is one of the first studies investigating the drivers of the sampling efforts for freshwater fish at a fine resolution but at continental and country scales [47]. The anthropogenic factors related to accessibility or to areas that are interesting to researchers previously explained the sampling efforts for terrestrial organisms [76] and, along with the environmental factors, also shaped the completeness pattern of the freshwater fish at different scales. The importance of the environmental conditions might also be related to the most common sampling procedure required for fish, which is normally carried out by professionals with sophisticated equipment (e.g., electrofishing) in locations where the conditions of the river (e.g., depth, water flow, or speed) allow the implementation of the sampling techniques [93]. Furthermore, freshwater biodiversity catalogues are less based on community observations due to the detection difficulties for these taxa. Thus, the sampling bias linked to human infrastructures was less relevant for fish than for other taxa [66].

Finally, our results can help to guide future survey efforts in those areas where the data are scarce, or they can help to design efficient sampling protocols [94]. The completeness map and the sampling-effort knowledge obtained in this study can be used to determine regions where the biodiversity data are insufficiently consistent [14] and can guide the selection of the locations for future surveys along with the multiple approaches increasingly being published to optimize sampling [92,95–97]. Moreover, the completeness map can be used as a bias proxy to correct or account for the survey efforts in ecological models [98,99], such as the SDMs, in order to report the uncertainty associated with the Wallacean shortfall in the modeling [100,101]. Future research steps regarding the sampling bias of freshwater fish should therefore be focused on filling the gaps in the undersampled regions, such as those of eastern Europe [102], with the potential for the discovery of new species [103] and

the investigation of how the sampling bias varies between the different taxonomic groups of stream fish [36].

## 5. Conclusions

Our study detected low freshwater fish inventory completeness in eastern Europe and recorded gaps in most European regions; better survey inventories are biased towards places with high accessibility but are highly dependent on the study country. Our findings might help to define optimum strategies for future sampling in Europe as well as to inform ecological models of the bias posed by the geographic data of the species in order to improve our knowledge of freshwater fish dynamics and therefore the conservation and management measures to be applied to one of the most endangered faunas in the world.

**Supplementary Materials:** The following supporting information can be downloaded at: https://www.mdpi.com/article/10.3390/fishes7060383/s1, Figure S1: Variograms showing low spatial autocorrelation for the GLMM residuals; Table S1. GBIF links to each country's database and their references; Table S2: Number of freshwater fish records and species and average completeness by country; Table S3: Eigenvalues showing the percentage of variances explained by each principal component for the PCA performed to group the 19 bioclimatic predictors; Table S4. Loading of each of the two factors resulting from the PCA explaining 69% of the climate information.

**Author Contributions:** M.R.-R. conceived the idea and compiled and curated the data; M.R.-R. and G.G. designed the methodology, analyzed the data, interpreted the outputs, contributed to the manuscript writing, gave approval for publication, and agreed to be accountable for any question related to this work. All authors have read and agreed to the published version of the manuscript.

**Funding:** This study was supported by the "Make Our Planet Great Again" initiative from Campus France to M.R.-R. Laboratoire Evolution et Diversité Biologique EDB lab was supported by 'Investissement d'Avenir' grants (CEBA, ref. ANR-10-LABX-0025; TULIP, ref. ANR-10-LABX-41).

**Institutional Review Board Statement:** Not applicable.

**Data Availability Statement:** The data used are derived from public domain resources, but all data and R scripts are available on request to M.R.R.

**Conflicts of Interest:** The authors have no conflict of interest.

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
