# Peer review of "Disentangling the Drivers of the Sampling Bias of Freshwater Fish across Europe"

_fishes, doi:10.3390/fishes7060383_

Round 1

Reviewer 1 Report

The study found that human infrastructure was not a critical driver for sampling efforts for freshwater fish taxa, and found an association between climate and freshwater sampling efforts.  However, the statistical tests should account for potential spatial autocorrelation.

The authors put some effort into obtaining data from national sources. But the data curation procedure should be better described when combining data from national sources with the Global Biodiversity Information Facility (GBIF).  What are the meta-data structures from each national source, and how were the national databases different from the GBIF? The inclusion criteria for the specie accumulative curve analyses might need to be adjusted based on the data quality from each national source.

What is the actual method behind the KnowR function? The method description sounded more like how to interpret these values, but another researcher won’t be able to repeat the analysis if the researcher is not familiar with the cited R package.  

Abstract

The last sentence of paragraph 3 “Accessibility …” is vague, I think the authors want to say that both accessibility parameters and climate factors are associated with the completeness indicator, if so, please clarify.

Lines 120-121: The sentence “Other national …” is vague, I think it means that requests for national databases were made to several other countries (undefined), but access was only granted from Slovenia.

Line 127: removing duplicates, this requires clarification, duplication can also occur when the same individuals were recorded multiple times. Simply rounding the geographical coordinates might underestimate species richness. For example, if a specie was observed once in spring, 2022, and once in fall, 2022, these two observations should not be duplicates, but will be treated as such in the current definition.

Line 174: there is a typo, stast should be stats.

Line 178: This model assumes a lack of spatial pseudo-replications across countries. This assumption can be tested within the GLMM framework using spatially structured random effects.

Line 262: “notorious” carries more subjectivity than necessary, can the authors use a more objective work to explain the effect of digitization?

Lines 308-310: the sentence “Sampling is also … conditions” seems to be circular reasoning, readers need to understand climate conditions in order to understand “common environment”, but neither is clearly defined.

Figure 1: how does the study differentiate between no data, and with data but with very low completeness? I find it confusing to mix these two and define them both as zero.

Reviewer 2 Report

Dear Authors,

Necessary minor corrections are indicated as sticky notes on the pdf. 

Best wishes

Author Response

We have followed all comments and suggestion from the sticky notes in the pdf (see new version)

Reviewer 3 Report

Dear editor,

The manuscript of Rodríguez-Rey and Grenouillet, that is considered for publication was very interesting, well written and scientifically sound. My suggestion is to accept it at the present form.

However, its layout does not mach with the journal's template and some extensive reformation is required.

Sincerely yours 

Author Response

We have make the required changes to meet the format required by the journal. Thanks for the review.

Round 2

Reviewer 1 Report

L20-21: "Accessibility...but also...", this sentence may be better stated in two sub-sentences.

L65-66: this sentence "However..." seems out of place in the paragraph, which focuses on the known bias due to accessibility.

L99-100: "were requested to the authors obtaining successful results for Slovenia..." this sentence seems to say that Slovenia granted the author's data request, but many other countries did not. Please clarify.

L101. GBIF has documentation regarding meta-data structure and allows documentation of geospatial issues. To replicate this analysis, the readers need to know how the authors determined which records to include, and the rationale for such inclusion criteria. This is an important issue.

L105: I maintain that de-duplication by rounding long/lat is likely significant in interpreting the results, especially those relating accessibility with the dynamic environmental predictors. Besides, the KnowB package advised against removing duplicate records from heterogeneous sources as well. See PP243, Paragraph 2 of [61]. Please justify your choice, maybe by comparing results with an alternative de-duplication procedure.

[61] Lobo, J.M.; Hortal, J.; Yela, J.L.; Millán, A.; Sánchez-Fernández, D.; García-Roselló, E.; González-Dacosta, J.; Heine, J.; González-442 Vilas, L.; Guisande, C. KnowBR: An application to map the geographical variation of survey effort and identify well-surveyed 443 areas from biodiversity databases. Ecological Indicators 2018, 91, 241-248.

L113: The spatial unit is a 10 x 10 km grid cell, not polygons, hence they are regular. It's natural to use KnowB for regular cells instead of polygons. Please explain.

L160: "Contrary" is an adjective? please use a different word.

L193-193: The authors conducted variogram analysis of the completeness and deviance residuals from a GLMM, and found no evidence of spatial autocorrelation. This sentence does not capture that.

Also, it's unclear why the authors cut off the distance of the variogram for data to only 50, but allows the variogram for the residual to go out to 4000. What are the spatial units for these variograms?  Did the authors fit a spatial GLMM to compare with the non-spatial GLMM?

Figure 1: I maintain my earlier request to use different colors for cells without record and cells with record and near zero estimated completeness. Several papers cited by the authors differentiate the missing values with the zeros, so this seems to be the standard practice.

Author Response

Our comments are in the attached response letter.
